# DO NOT START WITH TREMBLING HANDS: IMPROVING MULTI-AGENT REINFORCEMENT LEARNING WITH STABLE PREFIX POLICY

## ABSTRACT

In multi-agent reinforcement learning (MARL), the $\epsilon$-greedy method plays an important role in balancing exploration and exploitation during the decision-making process in value-based algorithms. However, the $\epsilon$-greedy algorithm can be deemed as the concept of "trembling hands" in game theory when the agents are more in need of exploitation, which may result in the Trembling Hands Nash Equilibrium solution, a suboptimal policy convergence. Besides, eliminating the $\epsilon$-greedy algorithm leaves no exploration and may lead to unacceptable local optimal policies. To address this dilemma, we use the previously collected trajectories to construct a Monte-Carlo Trajectory Tree, so that an existing optimal template, a sequence of state prototypes, can be planned out. The agents start by following the planned template and act according to the policy without exploration, **Stable Prefix Policy**. The agents will adaptively dropout and begin to explore by following the $\epsilon$-greedy method when the policy still needs exploration. We scale our approach to various value-based MARL methods and empirically verify our method in a cooperative MARL task, SMAC benchmarks. Experimental results demonstrate that our method achieves not only better performance but also faster convergence speed than baseline algorithms within 2M time steps.

## 1 INTRODUCTION

Recent research on multi-agent reinforcement learning (MARL) has a very wide range of applications in the real world such as autonomous vehicle teams (Cao et al., 2012) and sensor networks (Zhang & Lesser, 2011). A number of MARL methods have been proposed to improve either value decomposition (Sunehag et al., 2017; Rashid et al., 2018; 2020; Wang et al., 2020a) or cooperative exploration (Yang et al., 2020; Mahajan et al., 2019; Wang et al., 2020b), among which value-based MARL methods (Sunehag et al., 2017; Son et al., 2019; Wang et al., 2019b) have shown outstanding performance on challenging tasks. e.g. StarCraft II (Samvelyan et al., 2019).

Moreover, most of the value-based MARL algorithms use $\epsilon$-greedy (Sutton & Barto, 1998) method to balance exploration and exploitation by choosing the greedy action with a probability $1 - \epsilon$ or a random choice action otherwise. However, such schemes are decision-making processes with Trembling Hands in game theory, in which sub-optimal solutions are the Trembling Hand Perfect Nash Equilibrium (THPNE).

In order to explain this phenomenon, we show a typical matrix game as described in Figure 1. Two solutions (T, L) and (B, R) are the two Nash Equilibrium results where (B, R) is the global-optimal solution and (T, L) is the sub-optimal one. If player 1 applies $\epsilon$-greedy method with $\epsilon = 0.2$, choosing action L by 0.1 and action R by 0.9, the expectation for player 2 to choose T is $1.81$. The expectation of action B is $1.8$, which is smaller than that of choosing T. Therefore, player 2 will always choose T when player 1 makes decisions with trembling hands and player 2 always chooses L for the same reason. In this case, the solution will fall into (T, L), the THPNE, which is not optimal. In the mixing networks of MARL, the Q values of each

| | player 1 | |
|---|---|---|
| | L | R |
| T | ( 1 , 1) | (1.9 , 0) |
| B | ( 0 , 1.9) | ( 2 , 2) |

(player 2)

Figure 1: A matrix game showing sub-optimal solutions with trembling hands.

agent are often added into $Q_{tot}$. Therefore, the calculation of $Q_{tot}$ is inaccurate and the errors are cumulated and propagated through the transitions among a trajectory.

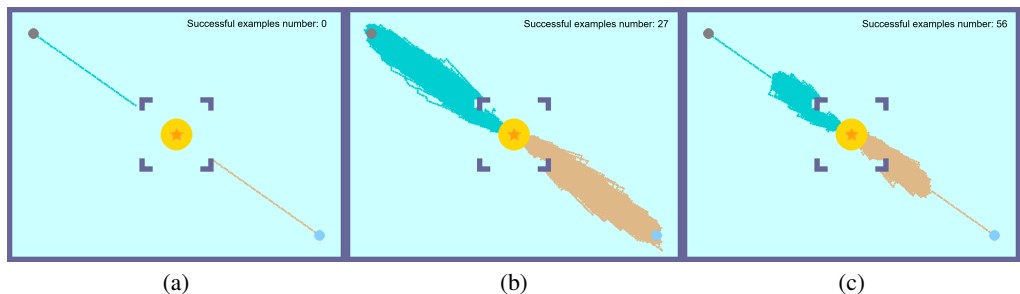

Figure 2: Two agents in the opposite are asked to reach the center goal **simultaneously**. Four obstacles around the central goal stop agents from reaching the goal. The default policy is strictly moving towards the center goal. The three figures above show the rollout traces that (a) agents choose actions greedily, (b) agents choose actions following $\epsilon$-greedy method through the rollout process, and (c) agents choose actions that start by stable prefix policy and follow $\epsilon$-greedy method in later time steps.

Figure 2 also shows the dilemma between the benefit of exploration and the sub-optimal solutions $\epsilon$-greedy method brings. Agents are trapped in local optima when greedy selections are applied only. When using $\epsilon$-greedy method, agents explore through the whole trajectories which makes them difficult to reach the goal. In contrast, if agents start with a stable policy for a few steps and apply $\epsilon$-greedy method afterward, the agents achieve a higher number of successful cases. Based on this, we propose **Stable Prefix Policy** (SPP) to encourage agents to follow the existing optimal trajectory planned from previously collected trajectory data. Specifically, we implement Monte-Carlo Trajectory Tree (MCT$^2$) to preserve the structure of previous trajectories. The existing optimal trajectory template planned from MCT$^2$ is used for guiding whether the agents are following the template during rollouts and assembling target values in the training process. When the agents dropout from the template, $\epsilon$-greedy method is activated afterwards.

The main contributions of this work are as follows: **1)** We propose the SPP method to rebalance the exploration and exploitation process when the policy of agents is close to the optimal policy during the training process. **2)** Our proposed method can be adapted to other value-based MARL algorithms with mixing networks with minor changes to existing MARL code-bases. **3)** We validate our methods empirically by extensive experiments on SMAC benchmarks. Experimental Results indicate that existing MARL methods equipped with our method can compete with or outperform original MARL methods in terms of the winning rates or cumulated rewards respectively within 2M time steps.

## 2 RELATED WORK

### 2.1 MULTI-AGENT REINFORCEMENT LEARNING.

In multi-agent value-based algorithms, the centralized value function, usually a joint Q-function, is decomposed into local utility functions. Many methods have been proposed to meet the Individual-Global-Maximum (IGM) (Bu et al., 2020) assumption, which indicates the consistency between the local optimal actions and the optimal global joint action. VDN (Lowe et al., 2017) and QMIX (Rashid et al., 2018) introduce additivity and monotonicity to Q-functions. QTRAN (Son et al., 2019) transforms IGM into optimization constraints. QPLEX (Wang et al., 2020a) uses duplex dueling network architecture to guarantee IGM assumption. Instead of focusing on value decomposition, multi-agent policy gradient algorithms provide a centralized value function to evaluate current joint policy and guide the update of each local utility network. Most policy-based MARL methods extend RL ideas, including MADDPG (Lowe et al., 2017), MATRPO (Foerster et al., 2017), MAPPO (Yu et al., 2022). FOP (Zhang et al., 2021) algorithm factorizes optimal joint policy by maximum entropy and MACPF (Wang et al., 2023) is the latest algorithm that mixes critic values of each agent.

## 2.2 EXPLORATION IN MULTI-AGENT REINFORCEMENT LEARNING.

Extended from single-agent reinforcement learning, the $\epsilon$-greedy method is widely applicable in value-based MARL algorithms. In this paper, our approach is based on the $\epsilon$-greedy exploration method and QMIX algorithm for reward credit allocation. In policy-based algorithms, such as MAPPO and COMA (Foerster et al., 2018), for exploration, multi-agent approaches rely on classical noise-based exploration in which agents explore local regions that are close to their individual actor policy. Another line of coordinated exploration algorithms has been proposed. Multi-agent variational exploration (MAVEN) (Mahajan et al., 2019) introduces a latent space for hierarchical control. Agents condition their behavior on the latent variable to perform committed exploration. Influence-based exploration (Wang et al., 2019a) captures the influence of one agent's behavior on others. Agents are encouraged to visit 'interaction points' that will change other agents' behavior.

## 3 BACKGROUNDS

A fully cooperative multi-agent task is described as a Dec-POMDP (Oliehoek et al., 2016) task which consists of a tuple $G = \langle S, A, P, r, Z, O, N, \gamma \rangle$ in which $s \in S$ is the true state of the environment and $N$ is the number of agents. At each time step, each agent $i \in N \equiv \{1, \ldots, n\}$ chooses an action $a_i \in A$ which forms the joint action $\mathbf{a} \in \mathbf{A} \equiv A^N$. The transition on the environment is according to the state transition function that $P(\cdot|s, \mathbf{a}) : S \times \mathbf{A} \times S \to [0, 1]$. The reward function, $r(s, \mathbf{a}) : S \times A \to \mathbb{R}$, is shared among all the agents, and $\gamma \in [0, 1)$ is the discount factor for future reward penalty.

Partially observable scenarios are considered in this paper that each agent draws individual observations $z \in Z$ of the environment according to the observation functions $O(s, i) : S \times N \to Z$. Meanwhile, the action-observation history, $\tau_i \in T \equiv (Z \times A)^*$, is preserved for each agent and conditions the stochastic policy $\pi_i(a_i|\tau_i) : T \times A \to [0, 1]$. The policy $\pi$ for each agent is determined by a joint action-value function: $Q^\pi(s^t, \mathbf{a}^t) = \mathbb{E}_{s^{t+1:\infty}, \mathbf{a}^{t+1:\infty}}[R^t|s^t, \mathbf{a}^t]$, in which the accumulated reward is considered as a discounted return and formulated as $R^t = \sum_{i=0}^{\infty} \gamma^i r^{t+i}$. After the rollout process, the whole trajectory from the initial transition to terminated transition $< (s_0, \mathbf{o}_0, \mathbf{a}_0, r_0), \ldots, (s_H, \mathbf{o}_H, \mathbf{a}_H, r_H) >$ are stored in the replay buffer.

Deep q-learning algorithm aims to find the optimal joint action-value function $Q^*(s, \mathbf{a}; \theta) = r(s, \mathbf{a}) + \gamma \mathbb{E}_{s'}[\max_{\mathbf{a}'} Q^*(s', \mathbf{a}'; \theta)]$. Due to partial observability, $Q(\tau, \mathbf{a}; \theta)$ is used in place of $Q(s, \mathbf{a}; \theta)$ and parameters $\theta$ are learnt by minimizing the expected TD error. Centralized training and decentralized execution (CTDE) enables agents to acquire global states during the training and only individual observations during the testing execution. In multi-agent settings, VDN learns a joint action-value function $Q_{tot}(\tau, \mathbf{a})$ as the sum of individual value functions: $Q_{tot}^{\text{VDN}}(\tau, \mathbf{a}) = \sum_{i=1}^{n} Q_i(\tau_i, a_i)$. QMIX introduces a monotonic restriction $\forall i \in \mathcal{N}, \frac{\partial Q_{tot}^{\text{QMIX}}(\tau, \mathbf{a})}{\partial Q_i(\tau_i, a_i)} > 0$ to the mixing network to meet the IGM assumption. IGM asserts the consistency between joint and local greedy action selections in the joint action-value $Q_{tot}(\tau, \mathbf{a})$ and individual action-values $[Q_i(\tau_i, a_i)]_{i=1}^{n}$:

$$\arg\max_{\mathbf{a} \in \mathcal{A}} Q_{tot}(\tau, \mathbf{a}) = \begin{pmatrix} \arg\max_{a_1 \in \mathcal{A}} Q_1(\tau_1, a_1) \\ \vdots \\ \arg\max_{a_n \in \mathcal{A}} Q_n(\tau_n, a_n) \end{pmatrix}.$$

## 4 METHOD

In this section, we introduce the overall architecture of our method and describe the generation of the stable prefix policy. Our method divides the decision-making of the existing MARL methods into two phases: our Stable Prefix Policy and vanilla policy. SPP balances the exploration and exploitation during the trajectory planning process with UCT, with Dirichlet noise during the planning phase. For planning, we establish a trajectory tree from data in the replay buffer in the Monte-Carlo Tree structure, which we call Monte-Carlo Trajectory Tree (MCT$^2$), to plan out the existing optimal trajectory. Instead of selecting one action from MCT planning, our work uses MCT to preserve trajectories across episodes to provide trajectory templates for utility network training. Additionally, we describe the rollout process, the target value assembling method, and the training pipeline in this section. The pseudo-code is provided in Appendix A.

### 4.1 ARCHITECTURE

The training process of value-based MARL algorithms is the $Q$ value Temporal Difference (TD) updating of each agent's utility network. In QMIX and the algorithms derived from QMIX, TD updates are applied to the mixed $Q_{tot}$ value. The utility network is composed of multi-layer perceptron (MLP) layers and Gate Recurrent Unit (GRU) cells in which $h_i^t$ is the historical hidden state. Similar to QMIX algorithm, the utility network at time step $t$ of agent $i$ takes the observation $o_i^t$ and its chosen action $a_i^t$ as an input and outputs the $Q_i(\tau_i, a_i)$ of each agent according to the encoded history state $\tau_i$. Then, these $Q$ values are fed into the mixing network which guarantees the monotonic constraints by hyper-networks and the $Q_{tot}(\tau, \mathbf{a})$ is used for TD learning.

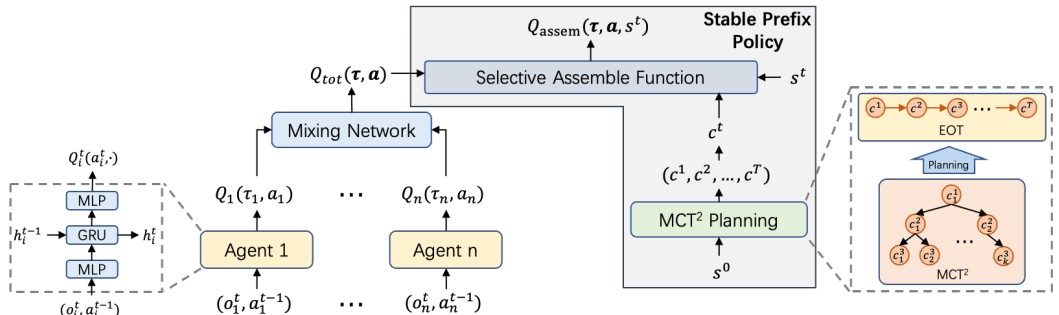

Figure 3: The utility networks and the mixing networks are from original MARL algorithms. Our method plans an existing optimal trajectory (EOT) by the trajectory tree (MCT$^2$). During the training, the selective assemble function assembles $Q_{tot}$ of each sample in a batch by comparing the template cluster $c^t$ with the input true state $s_t$. The $Q_{assem}$ is used for TD update.

As shown in Figure 3, our stable prefix policy is dependent on the time step $t$. To summarize the states into a few categories, we train a KMeans classifier $\phi(c|s)$ periodically by the data sampled from the replay buffer. To plan a potential optimal trajectory from MCT$^2$, the state $s^0$ (the initial state) is classified into a cluster $c^0$. Then the existing optimal trajectory is selected from the root node $c^0$ according to the probabilistic upper confidence bound (PUCB) value of each node and the sequence of cluster IDs is generated. At time step $t$, $c^t$ is used to be compared with the true state cluster and control the $Q$ assembling process.

Based on the trained classifier $\phi(c|s)$ and the sequence of transitions from the replay buffer during the training process, our classifier predicts the cluster ID of each state in each time step $t$. Whether the agents are following the trajectory template can be determined by the comparison between the predicted cluster IDs $\hat{c}^t$ (from $\phi(s^t)$) and the cluster from our stable prefix policy $c^t$. Once confirming the agents are following the template, the target value calculated by $Q_{tot}(\tau_\mathbf{n}, \mathbf{a_n})$ is assembled with other target values with the same cluster ID to calculate TD error. According to the CTDE settings, during the testing execution phase, the actions are conditioned only on the utility networks without SPP and without $\epsilon$-greedy exploration.

### 4.2 STABLE PREFIX POLICY

As shown in Figure 4, to plan out an existing optimal trajectory as a template from previously collected interactive data, our method generates trajectory trees by the data sampled from the replay buffer in Monte-Carlo Tree structure. We randomly select $t_{inter}$ trajectories and apply clustering methods to assign states $s$ into a cluster $c$ such that similar states can be assigned to the same cluster. A transition $(s^t, \mathbf{a^t}, s^{t+1}, r^{t+1})$ can be regarded as a visit from a node with cluster ID $\phi(s^t)$ to its child node with $\phi(s^{t+1})$. Meanwhile, the expectation rewards from one cluster of states to its subsequent clusters are stored in the tree. Apart from the IDs of clusters, the value of a node $v(n)$ is also stored in the node and is calculated by the following formula:

$$v(n) = \lambda v(n) + (1 - \lambda) \times \sum_{cn \in children(n)} \pi(cn|n) \times (R_{n \to cn} + \gamma v(cn)). \tag{1}$$

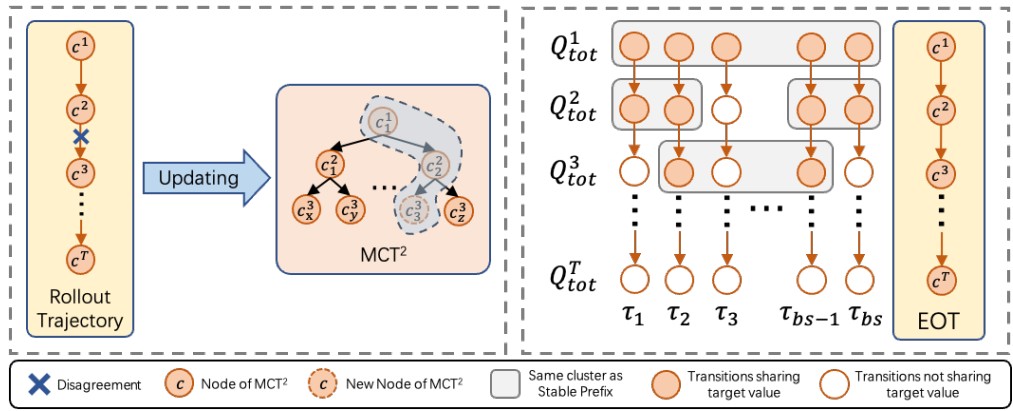

Figure 4: **Left:** Agents follow the template trajectory in the rollout process and encounter disagreement on cluster $c^3$. Then the transitions prior to $c^3$ are updated and a new node with $c^3$ is generated in our MCT². In the $c_i^j$ of MCT², $i$ is the time step of a state as well as the depth of a node and $j$ is the cluster ID. **Right:** A batch of transition sequences are converted to cluster transitions. For each transition in each time step, $Q_{tot}^t$ of nodes that have the same cluster IDs as the existing optimal trajectory are assembled. The transitions after the dropout process will not share the target value and the $Q_{tot}$ will not be assembled. For example, $Q_{tot}^3$ of $\tau_3$ is not assembled because $\tau_3$ has already dropped out in step 2.

In the formula above, $v(n)$ is the value of the node $n$ and $cn$ are the children nodes of the node $n$. $\pi(cn|n)$ is the probability of visiting a child node $cn$ from it parent node $n$ and is usually calculated by counting numbers. $R_{n \to cn}$ is the expectation of the historical rewards from node $n$ to node $cn$. $\gamma$ is the discount factor and $v(cn)$ is the value of child node $cn$. $\lambda$ is dynamically changed with the visit number of cluster $\Gamma_n$ and empirically defined as $1/\Gamma_n$.

During the establishment of the MCT², we follow the procedure in Efficient Zero (Ye et al., 2021). A newly selected node will be expanded with the average reward and policy as its prior. Additionally, when the root node is to expand, we apply the Dirichlet noise to the policy prior to give more explorations.

$$\pi(cn|n) := (1 - \rho)\pi(cn|n) + \rho\mathcal{N}_\mathcal{D}(\xi) \tag{2}$$

, where $\mathcal{N}_\mathcal{D}(\xi)$ is the Dirichlet noise distribution, $\rho$ and $\xi$ is set to 0.25 and 0.3. However, we do not use any nose and set $\rho$ to 0 for the non-root node or during evaluations.

Based on the MCT² implementation described above, we greedily select routes, a sequence of cluster IDs, from the root node to a leaf according to the PUCB values from each parent node $n$ to its child node $cn$. Inside the formula below, $c_{ucb}$ is the hyper-parameter for balancing exploration and exploitation.

$$c = \underset{cn \in children(n)}{\arg\max} \left( v(cn) + c_{ucb} \cdot \pi(cn|n) \cdot \sqrt{\left(\frac{\log \Gamma_{cn}}{1 + \Gamma_n}\right)} \right). \tag{3}$$

After the selection, an optimal path of clusters is selected from the root node to a leaf node, $(c^0, c^1, ...c^T)$ within time steps $T$, which will be used for training and rollout process.

During the rollout process, agents start by following the template trajectory generated by MCT². When the agents are following the template, the actions are selected greedily according to their $Q$ values for full exploitation. However, once the agents dropout from the template trajectory in a time step $t$ ($c^t \neq \phi(s^t)$), actions are generated by $\epsilon$-greedy for exploration in the latter rollout steps.

After the rollout processes, the trajectories from the environment interactions will be used to update the MCT². The states **s** are classified into clusters **c** which instead form the transition sequences $(c^0, a^0, c^1, a^1...c^T)$. The values of the node before the dropout time step will be updated or created in the MCT². It is worth noting that MCT² only concentrates on the cluster transitions **without** actions. In such a way, our stable prefix policy only focuses on the optimal subsequent states no matter what actions are taken by the agents.

### 4.3 TRAINING PIPELINE

The existing template trajectories are also used in the training process. A mini-batch of trajectories is sampled from the replay buffer to train the utility network. Our $\text{MCT}^2$ generates a template for each sampled trajectory to find the time step that the agents dropout from the template. As shown in Figure 3, the $Q_{tot}(\tau, a)$ are calculated from the mixing network and the cluster ID $c^t$ is the output of our stable prefix policy. The target values $y$ are calculated by:

$$y^t = r^t + \gamma[\mathbb{1}(c^{t+1} = \phi(s^{t+1})) \cdot Q_{assem}^{t+1}(s^{t+1}) + (1 - \mathbb{1}(c^{t+1} = \phi(s^{t+1}))) \cdot Q_{tot}(\tau, a^{t+1})] \quad (4)$$

While calculating the target $y$, we also assemble the target values of the same cluster node among the sampled batch of sequences such that the target values are close to the expectation of true discounted returns from that state.

$$Q_{assem}^t(s^t) = \frac{(\sum_{i=1}^{bs} Q_{tot}(\tau_i, a_i) \cdot \mathbb{1}(c^t = \phi(s^t)))}{\sum_{i=1}^{bs} \mathbb{1}(c^t = \phi(s^t))} \quad (5)$$

Inside the formula above, $bs$ is the batch size of sampled data, $Q_{tot}(\tau_i, a_i)$ is calculated by adding the rewards to the value of the subsequent node in the template, and the $c^t$ is the cluster node from the trajectory. Because our method takes $Q_{tot}$ and trajectory tree into consideration, our method can be adapted to other value-based MARL algorithms with mixing networks.

### 4.4 SAMPLE COMPLEXITY ANALYSIS

In this section, We linked our SPP method to the framework in Koenig & Simmons (1993), verifying that our SPP method can achieve a polynomial sample complexity. As we need to calculate the sample complexity of SPP method. Before that, since our SPP method uses the clustering method for feature extraction, we also need to give a reasonable assumption for the feature extraction module in our algorithm.

**Assumption 1** *Assume that the state is parametrized by some feature mapping (clustering mapping) such that for any policy $\pi$, $Q_{assem}$ and $\pi(s)$ depend only on $\phi(s)$, the stable prefix policy $\pi^{spp}$ cover the states visited by the optimal policy:*

$$\sup_{s,t} \frac{d_t^{\pi^*}(\phi(s))}{d_t^{\pi^{spp}}(\phi(s))} \leq C$$

Where $\pi^*$ is the optimal policy, $d_t^{\pi}$ is the state visit distribution under a policy $\pi$ in time step $t$, $\phi(\cdot)$ is a feature extractor of the policy, and the constant $C$ denotes an upper bound on the coverage ratioXie et al. (2022) between $\pi^{spp}$ and optimal policy $\pi^*$.

Assumption 1 indicates that the distributions of states being visited by each of the feature extractors corresponding to SPP $\pi^{spp}$ and utility policy $\pi$ should not be too different from each other. The ratio is sometimes called the distribution mismatch coefficient in the literature of policy gradient methods (Agarwal et al., 2021). We can show that given Assumption 1 our method explores the current time step without dropout of any state which gives good performance guarantees for MDP with general function approximation.

**Theorem 1 (Uchendu et al. (2023) theorem 4.3)** *With an appropriate choice of training and evaluation process, our approach in algorithm 1 guarantees a near-optimal bound up to factor of $C \times poly(H)$ for MDP with general function approximation.*

At this point, we have obtained all the results we need, showing that our SPP method achieves a polynomial sample complexity, providing a reasonable assumption 1 holds. Although polynomial or near optimal-bound can be achieved by many optimism-based methods (Jin et al., 2018; Ouyang et al., 2017), our approach further constructs a bonus for uncertainty, which improves the empirical performance of our SPP method.

# 5 EXPERIMENTS

We evaluate the performance of our method via the fully cooperative StarCraftII micro-management challenges by the mean winning rate in each scenario. In this environment, we mainly present 9 out of 23 scenarios with 3 levels of difficulty. Meanwhile, ablation studies are also presented to show the adaptability of our approach to other value-based algorithms, the influence of effective horizons, and the influence of cluster sizes during the training process. The details of other SMAC tasks are shown in Appendix B.

## 5.1 EXPERIMENT SETTINGS

**SMAC:** We verify our proposed stable prefix policy methods on 9 subtasks of three difficulties, a) simple tasks including 8m, 1c3s5z, and MMM, b) hard tasks including 3s_vs_5z, 5m_vs_96m, 2c_vs_64zg, and c) super-hard scenarios 3s5z_vs_3s6z, MMM2, and 6h_vs_8z. The difficulty is set as 7 by default. The winning rates of battles are calculated by the mean of 5 different seeds and smoothed by 0.8 for better visualization **within 2M time steps**.

**Baselines:** We adapt our method to QMIX and W-QMIX algorithms and compare our methods to the value-based QPLEX algorithm, popular policy-based algorithm MAPPO, and currently the latest actor-critic algorithm MACPF. In the ablation study, we also adapt our method to QPLEX algorithm. The QMIX, QPLEX, and W-QMIX in this paper are from pymarl codebase (Hu et al., 2021). MACPF is from the codebase (Zhang et al., 2021; Wang et al., 2023) and MAPPO is provided by Yu et al. (2022)

## 5.2 EXPERIMENT RESULTS

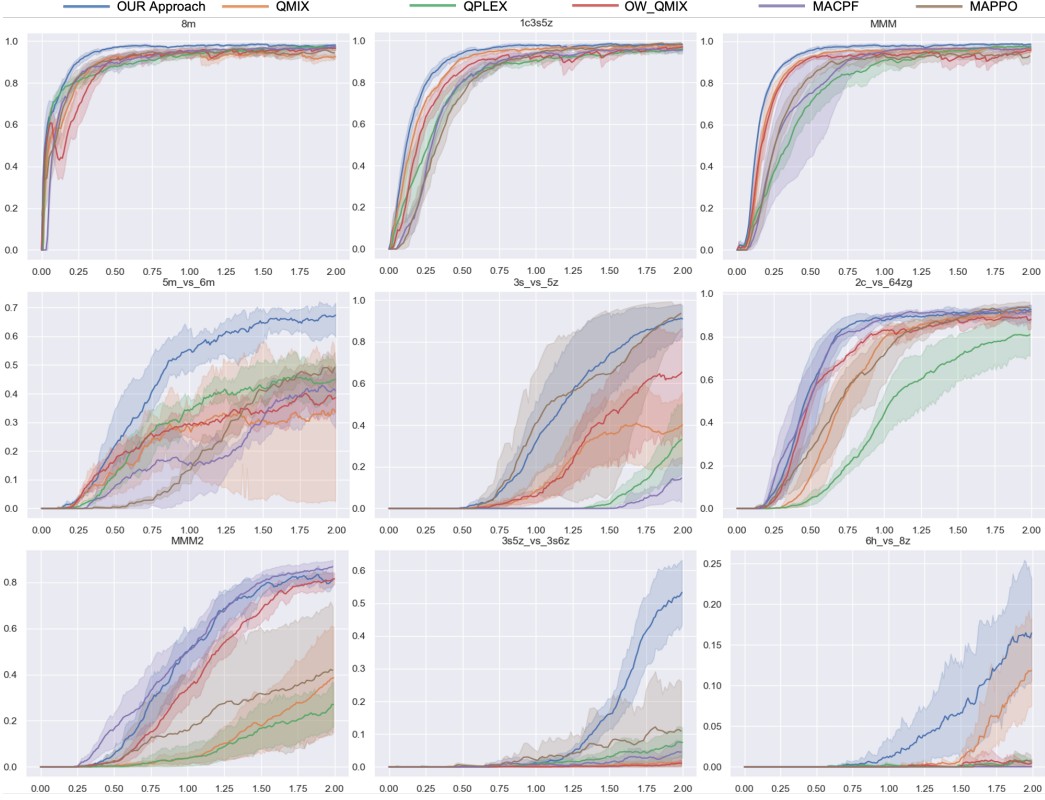

Figure 5: The winning rate curves evaluated on the nine SMAC tasks with three difficulties. The x-axis represents the time steps (1e6) being evaluated and the y-axis is the mean of the winning rate.

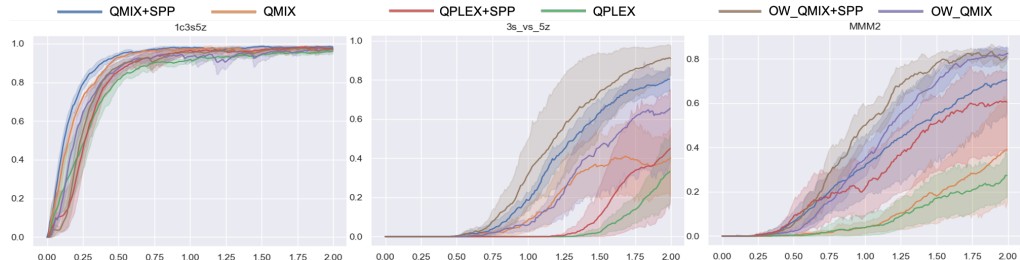

Figure 6: The winning rate curves evaluated on 1c3s5z (Easy), 3s_vs_5z (Hard), and MMM2 (Super-hard) scenarios. The x-axis is the time steps (1e6) that algorithms are evaluated at and the y-axis is the average value of the winning rate among 5 different seeds.

We mainly evaluate our proposed stable prefix policy method on QMIX algorithm on 9 benchmarks of SMAC, which is composed of three easy tasks, three hard tasks, and three super-hard tasks. To demonstrate the overall performance of each algorithm, Figure 5 plots the average test winning rate across the 9 scenarios. In 8m, 1c3s5z, and MMM tasks, our method outperforms other baselines but they almost achieve similar results due to the easy scenario. In hard tasks, including 5m_vs_6m and 2c_vs_64zg, and 3s_vs_5z, our proposed method can also compete with or outperform baseline algorithms. In the 3s_vs_5z scenario, our method has lower variance within 2M training steps. In the MMM2 task, our method can compete with policy-based methods, however, our proposed stable prefix policy still augments QMIX algorithm and outperforms other value-based methods. In the 6h_vs_8z and 3s5z_vs_3s6z tasks, not all the baselines show the winning rate and our method can achieve acceptable results. It is worth noting that we adjust the parameter size of the mixing network of QMIX and also apply both the original setting and the adjustment setting to other baselines. The better results of the two settings are shown in the graph. Other hyper-parameters are in Appendix C.

### 5.3 Ablation Studies

**Compatibility:** Because we implement a trajectory tree to provide current existing optimal trajectories for training and rollout and our stable prefix policy module is entirely on the basis of mixing networks, our method can be regarded as a plugin that can be adapted to other value-based MARL methods with minor changes. To test the compatibility of our work, we apply our method on QPLEX, and OW_QMIX algorithms in 1c3s5z, 3s_vs_5z, and MMM2 scenarios correspondingly.

According to Figure 6, in the 1c3s5z scenario and MMM2 task, both the QMIX with stable prefix and QPLEX with stable prefix outperform their original algorithms and OW_QMIX with stable prefix can compete with its origin. In the 3s_vs_5z scenario, all of the algorithms with our proposed stable prefix policy outperform the algorithms without prefix policy.

**Effectiveness:** During the rollout process, our proposed $MCT^2$ provides a potential optimal trajectory for agents to follow. Agents select actions according to their utility network and might encounter disagreements with the template in some time steps. Therefore, we record the portion of time steps that agents drop out from the template with the average length of an episode and analyze the influence of the dropout time step on the performance in three scenarios.

According to Figure 7 and the task specifications, 1c3s5z is an easy task for agents to focus fire on correct enemies, so agents have more probability to agree with the stable prefix trajectories. In the 3s_vs_5z task, agents should walk and attack, which is difficult for stable prefix policy to predict when to walk and attack. The important way to win MMM2 task is the control of Medivac and the ally to sacrifice, so the ratio of dropout length is high. According to the trend from the graph, as the policy network converges and the value of each node in our $MCT^2$ becomes accurate, the dropout ratio becomes higher in later training time steps.

## 6 Discussion

**Performance Enhancement:** According to the main experiment result in Figure 5, our method can compete with or outperform other baseline algorithms in most tasks. Our method can also outper-

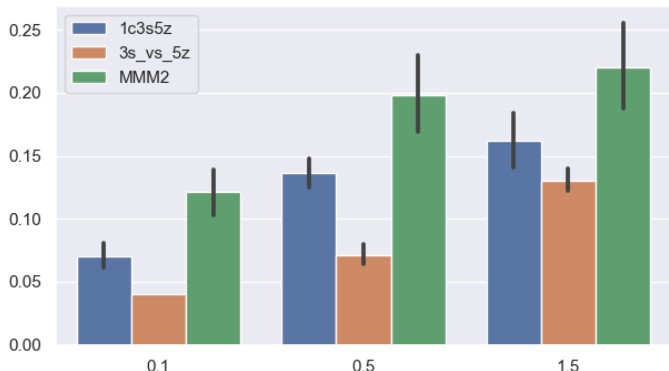

Figure 7: The drop out time step ratio of our QMIX+SPP algorithm on the 1c3s5z, 3s_vs_5z, and MMM2 scenarios in 100k, 500k, and 1500k time steps.

form other value-based algorithms in environments where policy-based algorithms are dominant. The existing optimal template trajectories provide agents currently the best route with the highest return. The $Q$ value assembling mechanism within a batch of trajectories reduces the error between the assembled $Q_{tot}$ and its true value.

**Adaptability:** We adapt our method on value-based MARL algorithms with mixing networks and assemble the $Q_{tot}$ for training. The essence of our work is providing a potential trajectory to agents and assembling a more accurate $Q$ value. Therefore, value-based MARL algorithms without centralized training, such as IQL, should also be suitable for our method. As for Actor-Critic MARL algorithms, the training of the critic modules is a value-based process, so our proposed method might be suitable for the critic training.

**Effectiveness:** We aim to find the optimal value without trembling hands when a sub-optimal policy can be obtained from historical interactions. Therefore, the stability of the prefix policy influences the dropout time step, the time step agents encounter disagreements with the provided template. According to Figure 7, the dropout time step is lower in the task where agents need to explore more during the early time steps. When the task is easy enough or agents do not need much exploration, the dropout time step will rise during the rollout process. In summary, the dropout time step is empirically positively correlated to training time steps and negatively correlated to the task difficulty.

**Limitation** Our proposed method restricts the early exploration process and allocates more exploration budgets for later time steps so that the underlying algorithm can explore more on later time steps which increases the opportunities to explore higher return states. However, in the tasks that need too much agents' exploration, in which agents cannot find a path to success in limited time steps, the agents drop out of the prefix template quite early such that our method cannot contribute to the overall training process. Further discussions are shown in Appendix D.

## 7 CONCLUSION AND FUTURE WORK

In this work, we consider the dilemma between the need for exploration and sub-optimal decision-making with trembling hands. To solve the problem, we propose a plugin that consists of a stable prefix trajectory provider, the Monte-Carlo Trajectory Tree, and a selective assemble function. We show that the usage of our stable prefix policy can improve MARL algorithms' performance when their utility network is close to optimal. SMAC experimental results indicate that our method can be adapted to any value-based MARL method in terms of implementation and offers significant improvements to value-based MARL methods. Trembling Hand is the exploration dilemma in value-based reinforcement learning, however, policy-based MARL algorithms generate actions from mix-strategy policies. In the future, we might focus on the exploration dilemma from mixed strategies in policy-based algorithms.

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

## A   PSEUDOCODE FOR OUR APPROACH

---

**Algorithm 1** MARL with stable prefix policy

---

1: Warm-up $t_{inter}$ episodes
2: Train classifier $\phi(c|s)$
3: Sample $t_{inter}$ trajectories from the replay buffer and generate MCT$^2$
4: **while** within the maximum number of time steps **do**
5:     Reset environment and obtain $s^0$
6:     Generate template trajectory $(c^0, c^1, ...c^n)$ by $\phi(s^0)$
7:     Rollout **without** $\epsilon$-greedy until $\phi(s^t) \neq c^t$
8:     Rollout **with** $\epsilon$-greedy until terminated
9:     Insert the rollout trajectory into the replay buffer and update MCT$^2$
10:    **if** Every training interval **then**
11:        Sample a batch of trajectories from the replay buffer and generate templates by $\phi(c|s)$
12:        Assemble Q values by comparing $\phi(s^{i,t})$ with $c^{i,t}$
13:        Train utility networks by TD error.
14:    **end if**
15:    **if** Every $t_{inter}$ episodes **then**
16:        Destroy MCT$^2$ and train a new classifier $\phi(c|s)$
17:        Reconstruct an MCT$^2$ by trajectories from the replay buffer
18:    **end if**
19: **end while**

---

## B   MORE EXPERIMENTS ON SMAC

Apart from the graphs shown in our main paper, we also operate experiments on other SMAC tasks. The results are shown in Table 1 and Graph 8. The table shows the winning rate on different SMAC tasks among 5 different seeds per task. The best performance is shown by bold font and the second best performance is shown with underline. We also generate replays of the tasks and some of the "SC2Replay"s are shown in the attachments, all the other replays can be performed through StarCraftII software downloaded from Battle.net.

## C   EXPERIMENTS HYPER-PARAMETERS

Most of the hyper-parameters used in this paper are the default parameters from the code-base pymarl. The corresponding important parameters of SMAC and algorithms are listed below.

The QMIX algorithm we use is from pymarl code base (Samvelyan et al., 2019), the QPLEX and OW_QMIX are from pymarl2 code base (Hu et al., 2021), the MAPPO algorithm is from the official code-base (Yu et al., 2021) and MACPF is from the open-sourced code from paper Wang et al. (2023). The detailed hyper-parameters are listed as Table 4 and the modified hyper-parameter for task 3s5z_vs_3s6z is shown in Table 3:

Apart from the hyper-parameters in pymarl codebase. The hyper-parameters of MAPPO algorithm are the default settings provided by the codebase. This codebase specifies corresponding hyper-parameters for each scenario. We change the total training time steps to 2M and the evaluation episodes to 6.

## D   MORE DISCUSSION ON SMAC TASKS

**The influence of cluster size** The number of clusters influences the establishment of the trajectory tree. A larger number of clusters provides more precise state classifications and larger computational consumption. Additionally, the number of states in one cluster also influences the precision of the assembled Q value. Therefore, we empirically record the test winning rates of different numbers of clusters in Figure 9.

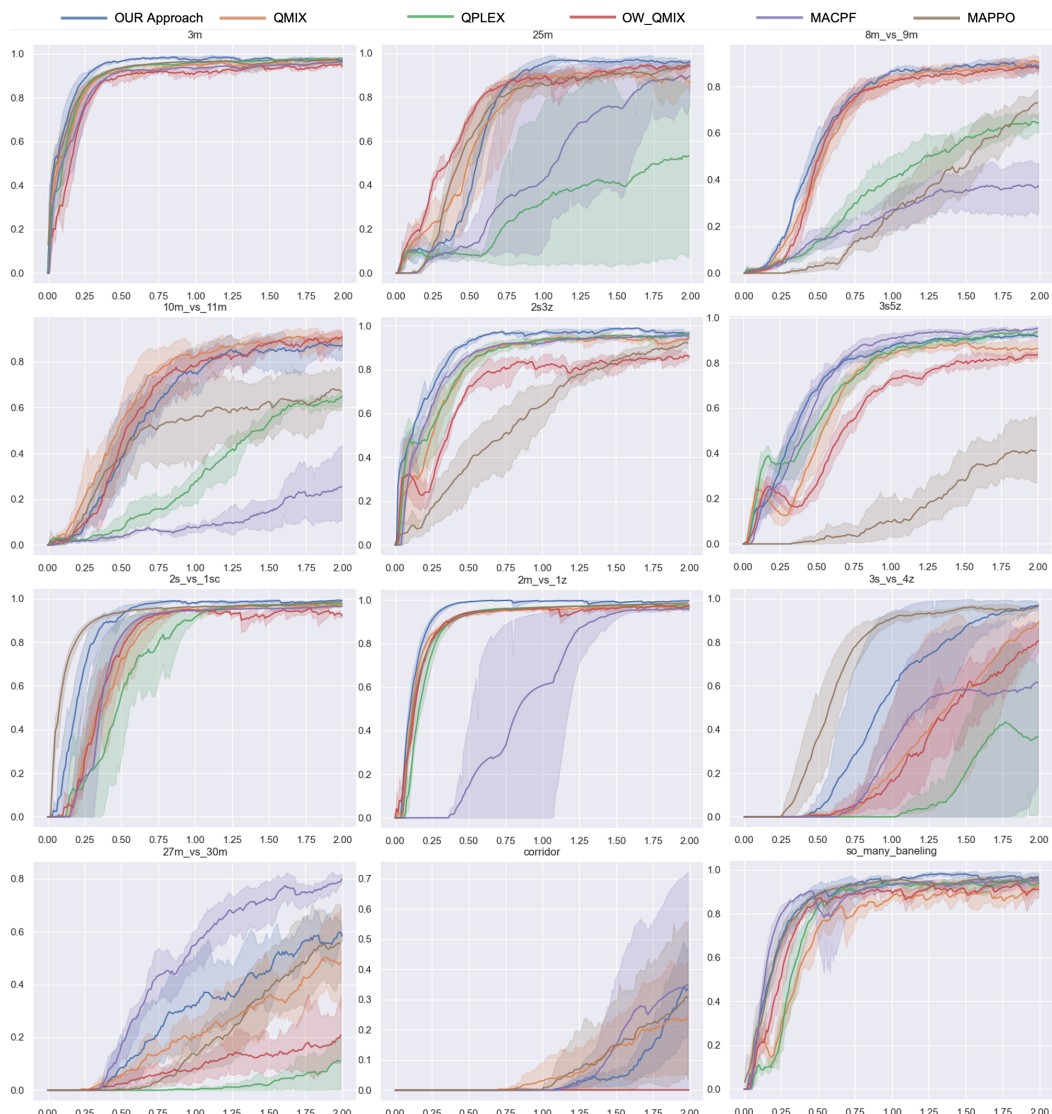

Figure 8: The winning rate curves evaluated on the 12 SMAC tasks with 5 different difficulties. The x-axis represents the time steps (1e6) being evaluated and the y-axis is the mean of the winning rate.

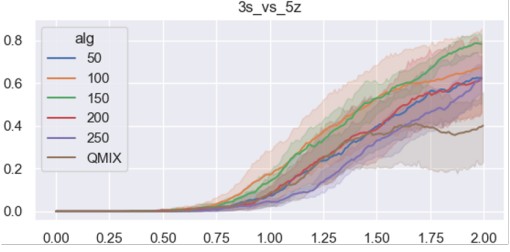

Figure 9: The testing winning rates of QMIX+SPP with different numbers of clusters and QMIX algorithm. The x-axis is the time steps (1e6) that algorithms are evaluated at and the y-axis is the average value of the winning rate.

According to the experimental results, the result of size 150 achieves the highest winning rate. A smaller number of clusters may classify states with larger variations into one group, which may

Table 1: Final performance in SMAC tasks

| tasks | Our Approach | QMIX | QPLEX | OW_QMIX | MAPPO | MACPF |
|---|---|---|---|---|---|---|
| 8m | **0.978** | 0.929 | 0.971 | 0.961 | 0.946 | 0.978 |
| 1c3s5z | 0.983 | 0.974 | 0.955 | 0.967 | **0.987** | 0.979 |
| MMM | 0.986 | 0.966 | 0.976 | 0.942 | 0.931 | **0.988** |
| 8m_vs_9m | 0.884 | **0.919** | 0.635 | 0.877 | 0.756 | 0.393 |
| 3s_vs_5z | 0.914 | 0.383 | 0.327 | 0.656 | **0.962** | 0.163 |
| 3s5z | 0.920 | 0.863 | 0.934 | 0.834 | 0.418 | **0.968** |
| 2c_vs_64zg | 0.931 | 0.922 | 0.823 | 0.901 | **0.954** | 0.945 |
| 3s5z_vs_3s6z | **0.538** | 0.015 | 0.074 | 0.009 | 0.110 | 0.061 |
| MMM2 | 0.823 | 0.382 | 0.263 | 0.808 | 0.436 | **0.898** |
| corridor | **0.430** | 0.250 | 0 | 0 | 0.330 | 0.374 |
| 3m | 0.989 | 0.981 | 0.988 | 0.960 | 0.989 | **0.994** |
| 25m | 0.954 | 0.872 | 0.530 | 0.949 | **0.969** | 0.930 |
| 5m_vs_6m | **0.670** | 0.339 | 0.445 | 0.387 | 0.495 | 0.445 |
| 10m_vs_11m | 0.873 | **0.921** | 0.645 | 0.906 | 0.702 | 0.271 |
| 27m_vs_30m | 0.586 | 0.512 | 0.105 | 0.191 | 0.582 | **0.817** |
| 2s3z | 0.965 | 0.948 | 0.974 | 0.871 | 0.933 | **0.985** |
| 3s_vs_3z | 0.979 | 0.972 | **0.992** | 0.967 | 0.987 | 0.982 |
| 3s_vs_4z | **0.965** | 0.892 | 0.368 | 0.812 | 0.962 | 0.632 |
| 2m_vs_1z | 0.995 | 0.980 | 0.986 | 0.985 | **0.999** | 0.992 |
| 6h_vs_8z | **0.170** | 0.119 | 0.008 | 0.006 | 0.001 | 0.010 |
| 2s_vs_1sc | 0.993 | 0.982 | 0.991 | 0.932 | **0.999** | 0.992 |
| so_many_baneling | 0.974 | 0.926 | 0.953 | 0.925 | 0.967 | **0.981** |

Table 2: SMAC task default settings

| | |
|---|---|
| continuing_episode | False |
| difficulty | 7 |
| move_amount | 2 |
| obs_all_heath | True |
| obs_last_action | False |
| obs_own_health | True |
| obs_terrain_height | False |
| reward_death_value | 10 |
| reward_defeat | 0 |
| reward_negative_scale | 0.5 |
| reward_only_positive | True |
| reward_scale | True |
| reward_scale_rate | 20 |
| reward_win | 200 |
| state_last_action | True |
| step_mul | 8 |
| heuristic_ai | False |

introduce extra bias when assembling the Q values. In contrast, a larger number of clusters slows down the convergence speed because of the lack of samples in a cluster when assembling Q values. Therefore, the cluster size should be balanced according to the size of the replay buffer and in this paper is set as $n_{state}/1000$, where the $n_{state}$ is the total number of states for training the classifier.

**Performance on tasks with single type of agents** According to the learning curves in Appendix B and main paper our method can compete with baseline algorithms without too much promotion in the tasks with single type of agents. The method for winning such a scenario is that agents should form a fan-shaped team formation and catch fire on enemies without damage overflows. In addition, agents should also move back when their health values are low. However, this strategy is easy to be

Table 3: Different hyper-parameters of 3s5z_vs_3s6z

| epsilon_start | 1.0 |
|---|---|
| epsilon_finish | 0.05 |
| epsilon_anneal_time | 100000 |
| batch_size | 128 |
| rnn_hidden_dim | 256 |
| hypernet_layers | 1 |
| hypernet_embed | 256 |
| optim | Adam |

Table 4: hyper-parameters for baseline algorithms

| parameter | QMIX+SPP | QMIX | QPLEX | OW_QMIX | MACPF |
|---|---|---|---|---|---|
| gamma | 0.99 | 0.99 | 0.99 | 0.99 | 0.99 |
| batch_size | 32 | 32 | 32 | 32 | 32 |
| buffer_size | 5000 | 5000 | 5000 | 5000 | 5000 |
| lr | 0.001 | 0.0005 | 0.0005 | 0.001 | 0.0005 |
| critic_lr | - | - | - | - | 0.0005 |
| optim_alpha | 0.99 | 0.99 | 0.99 | 0.99 | 0.99 |
| optim_eps | 0.00001 | 0.00001 | 0.00001 | 0.00001 | 0.00001 |
| rnn_hidden_dim | 64 | 64 | 64 | 64 | 64 |
| optim | RMSprop | RMSprop | RMSprop | RMSprop | RMSprop |
| action_selector | eps-greedy | eps-greedy | eps-greedy | eps-greedy | multinomial_seq |
| epsilon_start | 1.0 | 1.0 | 1.0 | 1.0 | 1.0 |
| epsilon_finish | 0.05 | 0.05 | 0.05 | 0.05 | 0.05 |
| epsilon_anneal_time | 50000 | 50000 | 50000 | 100000 | 50000 |
| agent_output_type | q | q | q | q | pi_logit |
| mixer | qmix | qmix | dmaq | qmix | dfop |
| mixing_embed_dim | 32 | 32 | 32 | 32 | 64 |
| hypernet_layers | 2 | 2 | - | 2 | - |
| hypernet_embed | 64 | 64 | 64 | 64 | 64 |
| adv_hypernet_layers | - | - | 3 | - | 1 |
| adv_hypernet_embed | - | - | 64 | - | 64 |
| td_lambda | 0.4 | 0.4 | 0.4 | 0.6 | 0.8 |
| double_q | True | False | True | True | False |
| num_kernel | - | - | 10 | - | - |
| is_minus_one | - | - | True | - | - |
| weighted_head | - | - | True | - | - |
| is_adv_attention | - | - | True | - | - |
| is_stop_gradient | - | - | True | - | - |
| central_mixing_embed_dim | - | - | - | 256 | - |
| central_action_embed | - | - | - | 1 | - |
| central_agent | - | - | - | central_rnn | - |
| central_rnn_hidden_dim | - | - | - | 64 | - |
| central_mixer | - | - | - | ff | - |
| n_head | - | - | - | - | 4 |
| attend_reg_coef | - | - | - | - | 0.001 |
| burn_in_period | - | - | - | - | 100 |
| dep_n_head | - | - | - | - | 4 |
| dep_embed_dim | - | - | - | - | 64 |
| dep_kv_dim | - | - | - | - | 64 |
| dep_output_dim | - | - | - | - | 64 |

explored and easy to be fulfilled in the former time steps. Therefore, most baseline algorithms and currently state-of-the-art algorithms have similar performances on theses scenarios.

**Performance on tasks with multi-type of agents** In the task with multi-type of agents, such as 3s5z, 1c3s5z, and MMM, our method outperforms other baseline algorithms. The conditions of winning the battle is much more complex. For example, one type of unit may counter another type of unit and some of agents should sacrifice themselves to be caught fire on for their allies to attack.

These strategies are difficult yet possible to be explored and agents may achieve high performance when following these strategy. In these scenarios, our method freezes the former policy for agents to follow and enables exploration afterwards. Therefore, in these scenarios, our method has significant improvements compared with other baseline algorithms.

**Performance on super-hard tasks** In super-hard tasks, baseline algorithms and currently state-of-the-art algorithms hardly have acceptable results. In the 6h_vs_8z scenario, none of the algorithms mentioned in this paper converges to optimal policy within 2M time steps. In the 3s5z_vs_3s6z scenario, we carefully adjust the hyper-parameters as shown in Table 3, which provides larger exploration opportunities to agents to find a path towards winning results. The agents' drop out length during the rollout process before finding a winning trajectory is quite short. The contribution of our work is finding an optimal solution without trembling hands when a sub-optimal solution can be found in historical interaction data. Without the sub-optimal paths, it is also difficult for our algorithm to achieve amazing results without carefully hyper-parameter adjustment.

According to Table 1, among 22 different tasks, our method achieves 6 best performances and 12 second best performances. The currently state-of-the-art algorithm, MACPF, achieves 7 best performances and 3 second best performances. However, some of the easy scenarios cannot distinguish the performances among all the scenarios. In the 5 super-hard subtasks including 2c_vs_64zg, 3s5z_vs_3s6z, MMM2, corridor, and 6h_vs_8z, our method achieves 3 best and 1 second best performances and MAPPO as well as MACPF achieves 1 best and 1 second best performances correspondingly, which indicates that our method can compete with and outperform sota actor-critic algorithms. As for value-based MARL baseline algorithms, our method achieves sota performances.

## E  HARDWARE FOR TRAINING

We operate our experiments on servers with 3.9 python version, AMD EPYC 7543 32-Core Processor CPU and NVIDIA GeForce RTX 3090 GPU. The maximum interaction time steps is 2.05M including test episodes and the StarCraftII version is 4.10. We set up 5 experiments with different seeds simultaneously and the actual time spent is about 6 hours and 30 minutes per task.

In our work, we cluster states into groups by KMeans provided by scikit-learn package, which needs extra time when training the classifier and constructing the trajectory tree. To deal with this problem, we install cupy and cuml (Raschka et al., 2020) which has compatible APIs with scikit-learn package. In this way, the classifier training and prediction process are calculated in GPU. Additionally, we also store the replay buffer inside GPU such that CPUs are fully responsible for running SMAC environment and GPUs are fully responsible for calculations.

## F  MPE

We also evaluate our method on three MPE (Terry et al., 2021) tasks, including spread, tag, and reference environment with 300k time steps. As shown in table 5, our method outperforms the baselines in the three tasks. However, little margin appeared in the performance on 300k time steps except tag environment. This result may indicate that MPE tasks are too simple and may not be challenging enough for a strong MARL algorithm.

Table 5: Average rewards per episode on three MPE tasks.

| Scenario | QMIX+SPP | QMIX | QPLEX | OW_QMIX | FOP | MACPF |
|---|---|---|---|---|---|---|
| Spread | **-92.662** | -95.133 | -95.212 | -93.691 | -96.182 | -92.974 |
| Tag | **248.99** | 112.202 | 195.294 | 237.19 | 205.13 | 242.74 |
| Reference | **-39.479** | -41.375 | -47.158 | -41.258 | -45.459 | -40.062 |

