# OpenReview forum: "Do not Start with Trembling Hands: Improving Multi-agent Reinforcement Learning with Stable Prefix Policy"
_ICLR.cc/2024/Conference — ICLR 2024 Conference Withdrawn Submission_

### Official Review · Reviewer_4uBa · 2023-10-29

**Soundness:** 2 fair
**Presentation:** 1 poor
**Contribution:** 2 fair
**Rating:** 5
**Confidence:** 4

**Summary:**

The paper addresses the problem of trembling hands in multi-agent systems, namely the negative effect that exploration has on the coordination between agents. This effect is particularly evident when $\espilon$-greedy policies are used as template policies. This work proposes a method to compute a template policy to be followed, instead of greedy policies, and offers some empirical evidence that the proposed method can be competitive on some experimental settings.

**Strengths:**

The trembling hands problem plays a central role in multi-agent coordination and the idea to follow specifically designed policies instead of more standard policies might lead to original results.

**Weaknesses:**

Unfortunately, the limitations of this work are extensive and I believe structural. The exposition is the main factor concerning this feedback, both from the point of view of the rhetorics and for the clarification of the original contributions. Here is a (non-exhaustive) list of points:
- In the abstract it is claimed that [you] " find that $\epsilon$- greedy policies can be deemed...", it is unclear how and why this was not already known. In its second part unclear. How do you compute such policies? What do you mean by "plan an existing optimal policy"? The description of what was done is unclear to me, and how this was done is absent.
- The related works section addresses the background rather than the related works, and the background is insufficient in the exposition to provide tools to understand what will be done later. Trembling Hands Nash Equilibria are never defined, for example. This leads to the fact that in the proposed method, it was unclear to me what portions of the whole regime are proper contributions of the work and what are not.
- The Theoretical Analysis is absent, meaning that in the way it is done is mostly unclear what it should suggest.
- The Experimental Evaluation suggests some cases of competitiveness but does not compare the methods from a computational point of view, which I believe would help understand the pros and cons of the proposed method. Finally, it was not clear to me how the hyper-optimization of the Sota algorithms used as baselines was done, both in the standard case and in the SDD-augmented case.
- A scientific analysis of the limitations would be needed.

Finally, some English phrasing is wrong and some typos are present (for example there should be an $\epsilon$ at the 9th line of the first page I believe)

**Questions:**

Unfortunately, the limitations seem extensive, and I believe a refactoring of the work is needed, I hope the comments suggest the portions of the work to be addressed, but I am open to further provide insights and discuss.

---

> ### Author Response · Authors · 2023-11-21
> **Respond to Weaknesses**
>
> Thank you for your comments and suggestions. After reading the summary as well as the weaknesses, I think there might be some gaps between your understanding of our paper and the contents we want to express. From my perspective, you are interested in the topic that how our approach solves the trembling hands' problems. However, we find the $\epsilon$-greedy methods in MARL are similar to the THPNE problem in game theory. We aim to solve the exploration-exploitation dilemma taken by the epsilon-greedy algorithm. Therefore, the paper is about the sequence of MARL problems instead of the one-step THPNE problem.  Despite of that, I will try my best to respond to your comments.
>
> > In the abstract it is claimed that [you] " find that $\epsilon$- greedy policies can be deemed...", it is unclear how and why this was not already known. In its second part unclear. How do you compute such policies?  What do you mean by "plan an existing optimal policy"?
>
> Based on the $\epsilon$-greedy algorithm, agents choose the best action with the probability of 1-$\epsilon$ and a random action otherwise. In game theory, players who make decisions with trembling hands will choose actions greedily with a probability of $p$ and other (random) actions with $1-p$. In such a way, the $\epsilon$-greedy method and the trembling hands' decision-making are correlated. Meanwhile [1] also states the relationship between $\epsilon$-greedy and the trembling hands process.
>
> In this paper, the stable prefix policy can be considered as a trajectory template. We establish a trajectory tree based on multiple historical interactive trajectories. In each episode, agents are in the initial state at the first time step. The trajectory tree will generate a potential optimal route of states from that initial state by greedily selecting the subsequent states with the largest value, which is the process of 'plan an existing optimal policy'.
>
> > The related works section addresses the background rather than the related works, and the background is insufficient in the exposition to provide tools to understand what will be done later.
>
> In this paper, we are focusing on the MARL exploration problem rather than the solution to the THPNE problem. Thus, we introduce MARL and some NE definitions in the Related Work section and introduce MARL basic notations and settings in the Background section. These contents might be sufficient for introducing the subsequent MARL methods.
>
> > The Theoretical Analysis is absent, meaning that in the way it is done is mostly unclear what it should suggest.
>
> We provide some theoretical analysis of the convergence guarantee and computational complexity of our algorithm in section 4.4. The THPNE problem is used for introducing the exploration dilemma, so we did not provide the theoretical analysis of the THPNE, which, we think, is not quite relevant to MARL problems in this paper.
>
> > it was not clear to me how the hyper-optimization of the Sota algorithms used as baselines was done, both in the standard case and in the SDD-augmented case.
>
> These Sota algorithms aim at solving the MARL problem instead of the THPNE problem, so none of them introduce their methods to solve that. Based on their algorithms, our approach does not focus on the problem either.
>
> > some English phrasing is wrong and some typos are present (for example there should be an $\epsilon$ at the 9th line of the first page I believe)
>
> We will double-check the expression methods and typos in this paper. However, at the 9th line of the first page, we want to express that agents choose the best actions with the largest values. Thus, agents choose actions greedily and there should not be an epsilon here.
>
> In the remaining time we would like to address the remaining concerns that you have that prevent you from accepting the paper. We would appreciate it if you could engage with us on the remaining concerns, as we want to address them.
>
>
>
> [1] Lee, Kiyeob, et al. "Reinforcement learning for mean field games with strategic complementarities." International Conference on Artificial Intelligence and Statistics.

---

> ### Comment · Reviewer_4uBa · 2023-11-22
>
> I would like to thank the Authors for the clarifications. I will proceed by points. What I was trying to point out is that the exposition of the work is the main concerning point to me, and I hope my comments will help:
> - In the abstract it is claimed that [you] " find that - greedy policies can be deemed...", it is unclear how and why this was not already known. In its second part unclear. How do you compute such policies? What do you mean by "plan an existing optimal policy"?
> In general what I was trying to say is that the abstract is not clear in what will be done, and it is imprecise in some parts as well. As an example of the general, unprecise exposition, I commented that in the abstract it is said that "[you] find that epsilon-greedy can be deemed as the concept ...". As you suggested as well, this result was not "found" in this work, it is a known fact. This is not a big issue, but the comment was to suggest that the phrasing of the work might be substantially revisited.
> - The related works section addresses the background rather than the related works, and the background is insufficient in the exposition to provide tools to understand what will be done later.
> I agree with the authors saying that "[they] are focusing on the MARL exploration problem rather than the solution to the THPNE problem". For this reason indeed, I suggested inserting in the related works some comments on how the problem of exploration in MARL is managed, rather than a few lines on MARL algorithms that do not tackle the problem of exploration, and a section on the Trembling Hands Problem, that is not the focus of the paper. In this sense, the related works section is rather unrelated to the work.
> - The Theoretical Analysis is absent, meaning that in the way it is done is mostly unclear what it should suggest.
> Indeed a theoretical analysis of the THPNE was not requested at all. What I was trying to say is that:
> Theorem 1 is by [2]. Assumption 1 is not discussed, how likely is it to be verified? Theorem 2 is by [1]
> What I mean is that the authors should better discuss how the proposed approach maps to pre-existing theoretical results, otherwise this section is rather poor and unjustified as a "Theoretical Analysis".
>
> -  some English phrasing is wrong and some typos are present (for example there should be an at the 9th line of the first page I believe).
> I am still a little confused here, mostly by the fact that the revised version has indeed $1-\epsilon$, so I really don't understand the answer "there should not be an epsilon here".
>
> Generally, then, I think the global exposition of the work still hinders the overall quality, and I don't see substantial changes done to address that. Nonetheless, I am positive about raising my score in view of the other's comments.
>
>
> [1] Koenig & Simmons (1993)
> [2] Uchendu et al. (2023)

---

> > ### Author Response · Authors · 2023-11-23
> > **Clarifications of the four concerns**
> >
> > Thank you for your comments and suggestions and for raising the score. After reading these comments, I realized that I misunderstood some of the comments above. Focusing on the four concerns, we will try our best to respond and hopefully address your concerns.
> >
> > > In the abstract it is claimed that [you] " find that - greedy policies can be deemed...", it is unclear how and why this was not already known. In its second part unclear. How do you compute such policies? What do you mean by "plan an existing optimal policy"?
> >
> > In the abstract of the previous version, the expression 'we found that the $\epsilon$-greedy policies can be deemed...' is overclaiming. Thus, we delete the 'we found that' and rephrase it to 'The $\epsilon$-greedy policies can be deemed as ...' in the latest revised version. Meanwhile, we clarify that we use the previously collected trajectories to **construct a Monte-Carlo Trajectory Tree** and plan an existing optimal template (a sequence of state prototypes) from the MCT$^2$ according to the largest value of PUCB calculation. When the agents are in the state prototype which is the same as the template (agents are following the template), the agents select actions according to the policy from the utility network without $\epsilon$-greedy exploration, which we call **Stable Prefix Policy**. In other words, the SPP is the policy generated from the utility network without exploration. Meanwhile, when to start the exploration process is determined by the 'whether agents are following the template'. If the agents arrive at a state that is not inside the potential optimal template, agents start to explore from this point.  Hopefully, we have explained the process clearly.
> >
> > > The related works section addresses the background rather than the related works, and the background is insufficient in the exposition to provide tools to understand what will be done later.
> >
> > In the related work section, we introduce several MARL algorithms and some works of Trembling Hands. The Trembling Hands problem is not focused in this paper,  so we delete this subsection and replace it by introducing the exploration methods in MARL algorithms, including $\epsilon$-greedy, distributional noises, coordinated exploration, and influence-based exploration. In this paper, our approach is based on the $\epsilon$-greedy exploration method and QMIX algorithm for reward credit allocation. We have already updated it in the latest revision version.
> >
> > > The Theoretical Analysis is absent, meaning that in the way it is done is mostly unclear what it should suggest.
> >
> > In the theoretical analysis section, we want to express the point that the vanilla $\epsilon$-greedy exploration method suffers from a sample complexity that is exponential in horizon $H $ (the length of an episode) for episode MDP, and our SPP method may achieve a lower polynomial sample complexity. The difference in the sample complexity contributes to the faster convergence of our approach and higher performance within early time steps.
> >
> > In such a way, Theorem 1 states that the sample complexity is exponential in the horizon to find a policy with sub-optimality smaller than 0.5. Then Assumption 1 introduces how the sub-optimality is evaluated. To make sub-optimality smaller, the distributions of states being visited by each of the feature extractors corresponding to SPP $\pi^{spp}$ and utility policy $\pi$ should not be too different from each other. Given assumption 1, our method explores the current time step without dropout of any state which gives good performance guarantees for MDP with general function approximation. Then theorem 2 indicates that the two-phase decision-making process guarantees a near-optimal bound up to a factor of $C \times poly(H)$.
> >
> > > Some English phrasing is wrong and some typos are present (for example there should be an at the 9th line of the first page I believe)
> >
> > Sorry for our misunderstanding on this comment. In the previous version, during the introduction of the $\epsilon$-greedy method, we used $1-p$ as the probability of choosing the best action. We realized that the $p$ is identical to the $\epsilon$ in this setting, so we have changed it to $1-\epsilon$ in the latest version.
> >
> > In the remaining time, we would like to address the remaining concerns that you have that prevent you from accepting the paper. We would appreciate it if you could engage with us on the remaining concerns, as we want to address them.

---

> > > ### Comment · Reviewer_4uBa · 2023-11-23
> > > **Some Final Comments**
> > >
> > > First of all, I am glad that the initial misunderstanding was solved. I believe the concerns I raised were mostly solved.
> > > I am aware that this work has more of an experimental contribution, but I would propose the following modifications to the "Theoretical Analysis" Section to reach what I believe would be more of a sound statement.
> > >
> > > What the authors show is that:
> > >  - there exists an MDP instance with exponential sample complexity when $\epsilon$-greedy is used.
> > >  - provided the Assumption 1, Theorem 2 could be applied to show polynomial sample complexity upper bound.
> > >
> > > What I suggest for the current version is to just use the second portion, namely :
> > > - provided the Assumption 1, Theorem 2 could be applied to show polynomial sample complexity upper bound and explain what in your setting corresponds to the " appropriate choice of training and evaluation process", as it was done in the original paper.
> > >
> > > The reason to avoid the first claim is that the link between a lower bound for $\epsilon$-greedy and an upper bound of the proposed method is vacuous since you are comparing different algorithms. $\epsilon$-greedy has exponential lower bounds yet it does not require any assumption, and I am not confident excluding that there might exist another MDP where your assumption is not satisfied and the exponentiality returns
> > >
> > > Generally, then, I would call the section "Sample Complexity Analysis", since it is too limited to be called a Theoretical Analysis.
> > >
> > >
> > > More generally:
> > >  - avoid claiming that "[you] showed that", since this would be an over-statement. The correct phrasing would be something like  "We linked our algorithm to the framework in .."
> > > - when using another work's theorem, please use the format "Theorem i [ Cite, theorem k]" where the first number is in your ordering and the second links to the theorem in the original paper. This is needed for theoretical reproducibility.

---

> > > > ### Author Response · Authors · 2023-11-23
> > > > **Section 'Sample Complexity Analysis'  rephrased**
> > > >
> > > > Thank you for your comments and suggestions. We refactor the section 'Theory Analysis' as 'Sample Complexity Analysis' as suggested. Meanwhile, we have updated our paper following the second portion and cite the theorem as 'Theorem 1 (Uchendu et al. (2023) theorem 4.3) '. Additionally, we revised our paper and rephrased the expression 'we show the SPP...' to 'we link our SPP method to the framework in Koenig & Simmons (1993)'. The newly rephrased contents in the section 'Sample Complexity Analysis' are marked in red in the new revision.
> > > >
> > > > Thank you again for your constructive feedback.

---

> ### Comment · Reviewer_4uBa · 2023-11-23
> **Comments for the Area Chair**
>
> Provided that the Authors refactor the section "Theoretical Analysis" as suggested, most of my concerns have been solved but I think the is still room for improvement. I am mostly worried about other's reviewer's doubts about the empirical corroboration, which can be shared, but I think I would need to know Reviewer RkkR's impression to give a final judgment.

---

### Official Review · Reviewer_3kTw · 2023-10-30

**Soundness:** 2 fair
**Presentation:** 2 fair
**Contribution:** 3 good
**Rating:** 5
**Confidence:** 4

**Summary:**

In order to alleviate the Trembling Hands Nash Equilibrium solution caused by the $\varepsilon$-greedy method in multi-agent reinforcement learning, this paper proposes a Stable Prefix Policy (SPP). SPP can rebalance the exploration and exploitation process when the policy of agents is close to the optimal policy during the training process. The specific method is to implement a Monte-Carlo Trajectory Tree (MCT$^2$) to preserve the structure of previous trajectories, which can plan the existing optimal trajectory template. When agents follow this template during rollouts, the target value is assembled with other target values with the same trajectories. When the agents drop out from the template, the $\varepsilon$-greedy method is activated afterward. SPP can be applied to any value decomposition framework, and experimental results in SMAC and MPE show that it can improve the performance of the basic algorithm.

**Strengths:**

1. This paper introduces the concept of the trembling hands into cooperative multi-agent reinforcement learning, which is reasonable and novel. The two didactic tasks in the introduction section fully demonstrate that the Trembling Hand Perfect Nash Equilibrium does exist in multi-agent tasks, which provides sufficient reasons for the proposal of the Stable Prefix Policy.
2. This paper implements MCT$^2$, which can plan an existing optimal trajectory (EOT) based on the trajectories in the replay buffer. SPP calculates the target value for TD update by comparing the actual trajectory of the agent with EOT, which is indeed a very novel approach.
3. Key resources (proofs, code, and replay videos) are available, and sufficient details are described such that an expert should be able to reproduce the main results.
4. The experimental results are thoroughly analyzed. For example, The dropout time step ratio in Figure 7 illustrates the working mechanism of SPP and is intuitive.

**Weaknesses:**

1. The proposed method is based on the premise that agents should be capable of finding a policy toward success from historical interactions. In other words, SPP relies heavily on the performance of the underlying algorithm.
2. The trembling hands is a concept in multi-agent games, but this paper only provides solutions in cooperative scenarios (Dec-POMDP problems). At the same time, SPP is only applied to value decomposition methods.
3. MCT$^2$ introduces more hyperparameters, which increases the difficulty and workload of hyperparameter tuning.
4. The proposed method was only evaluated on SMAC (the description of the experimental results in MPE is skimpy and unconvincing). SMAC is a popular multi-agent experimental platform but has been pointed out to have many shortcomings [1]. More and more researchers in the MARL community advocate conducting experiments in multiple different domains to evaluate the proposed algorithm comprehensively [2].

**Reference**

[1] Ellis, Benjamin et al. SMACv2: An Improved Benchmark for Cooperative Multi-Agent Reinforcement Learning. 2022.

[2] Gorsane, R. et al. Towards a Standardised Performance Evaluation Protocol for Cooperative MARL. 2022.

**Questions:**

1. What is the value of the hyperparameter $t_{inter}$? How does its value affect the performance?
2. The target value $y_t$ in vanilla QMIX is $y_t = r_t+\gamma\max_{a^{t+1}}Q_{tot}(s^{t+1}, a^{t+1} )$, which is related to $s^{t+1}$. Why is $y^t$ still related to $s^t$ in Eq. (3)?
3. Is there any theoretical basis to prove that $Q^t_{assem}$ is more accurate than the original $Q_{tot}$?
4. I think that in some scenarios, the SPP variant may be more likely to fall into a local optimal solution. Suppose that in such a scenario, agents can easily access the state corresponding to the suboptimal solution, while the state corresponding to the global optimal solution is in the opposite direction and relatively difficult to access (for example, further away from the initial position of the agents). The SPP variant may directly give up early exploration and find it difficult to converge to the global optimal solution. Of course, the above issue can be alleviated by adjusting $c_{ucb}$, but this requires sufficient prior knowledge.

---

> ### Author Response · Authors · 2023-11-21
> **Respond to Weaknesses**
>
> We thank the reviewer for the detailed review of our paper. We are motivated that you found the new perspective interesting and inspiring. After reading the reviews, we address your concerns below. Please let us know if further clarification is needed.
>
> > The proposed method is based on the premise that agents should be capable of finding a policy toward success from historical interactions. In other words, SPP relies heavily on the performance of the underlying algorithm.
>
> Our SPP approach divides the decision-making process into two stages: pre-decision based on planning and post-decision with action noise. Where the pre-decision is given through UCT planning, it is not necessarily required that the historical interactions contain a successful policy, which means that even if there are no successful policies in the historical trajectory, UCT planning is still able to select the regions with the highest value or exploration potential at the moment based on the existing interactions. In contrast to relying heavily on the base policy, our SPP approach takes advantage of this favorable property of UCT planning and instead improves the performance of the base policy. Our experimental results on SMAC benchmarks can fully validate this.
>
> > The trembling hand is a concept in multi-agent games, but this paper only provides solutions in cooperative scenarios (Dec-POMDP problems). At the same time, SPP is only applied to value decomposition methods.
>
> Thank you very much for your comments. For the perspective of better demonstrating the performance improvement of our SPP method compared to the original MARL method, we choose to test our method on the most popular benchmarks with MARL algorithms in the current MARL community, which mostly require computing the joint utility function to reach a better equilibrium of the game, and thus focus on solving the problem of credit allocation for cooperative MARL. Technically, our SPP method works for those scenarios with suboptimal equilibria due to trembling hands. We will actively look for MARL benchmarks other than cooperative scenarios for testing and welcome your further suggestions.
>
> > SMAC is a popular multi-agent experimental platform but has been pointed out to have many shortcomings [1]. More and more researchers in the MARL community advocate conducting experiments in multiple domains to evaluate the proposed algorithm comprehensively [2].
>
> We also fully agree with this comment that the SMACv1 environment lacks randomness in initializing the task settings. SMACv2 in contrast provides the configuration settings including the number of agents and enemies, the initialized positions, and the probability of generating an agent type. However, as far as we are concerned, this configuration provides too much randomness (I'd like to call it unstable). We believe SMACv2 is more suitable to test the migration capability and few-shot scenario adaptation ability of an algorithm. Despite that, we tested our approach on three scenarios of the SMACv2 environment. The items in the table are the average winning rate among 5 seeds and their variance.
>
> | 5_vs_5       | 0.5M           | 1M             | 1.5M           | 2M             |
> | ------------ | -------------- | -------------- | -------------- | -------------- |
> | MAPPO        | 8,67$\pm$4.82  | 21.39$\pm$7.91 | 28.77$\pm$6.56 | 35.94$\pm$9.24 |
> | QMIX         | 20.15$\pm$4.59 | 32.68$\pm$5.80 | 41.48$\pm$5.74 | 50.21$\pm$3.12 |
> | Our Approach | 19.07$\pm$5.62 | 38.97$\pm$9.32 | 45.11$\pm$8.34 | 52.24$\pm$6.73 |
>
> | 10_vs_10     | 0.5M           | 1M             | 1.5M            | 2M              |
> | ------------ | -------------- | -------------- | --------------- | --------------- |
> | MAPPO        | 3.54$\pm$2.50  | 11.84$\pm$1.85 | 16.98$\pm$2.18  | 30.85$\pm$7.77  |
> | QMIX         | 11.09$\pm$2.24 | 20.56$\pm$4.48 | 31.54$\pm$6.75  | 42.29$\pm$12.41 |
> | Our Approach | 6.70$\pm$2.87  | 24.48$\pm$8.78 | 37.75$\pm$10.35 | 43.45$\pm$9.80  |
>
> | 10_vs_11     | 0.5M          | 1M            | 1.5M          | 2M             |
> | ------------ | ------------- | ------------- | ------------- | -------------- |
> | MAPPO        | 0.76$\pm$0    | 0.83$\pm$0.91 | 0.74$\pm$0.47 | 1.84$\pm$1.29  |
> | QMIX         | 0.57$\pm$0.44 | 4.29$\pm$2.61 | 11.6$\pm$3.62 | 14.95$\pm$2.05 |
> | Our Approach | 0.34$\pm$0.75 | 4.51$\pm$1.92 | 10.3$\pm$4.54 | 12.67$\pm$5.26 |
>
> According to the results, our proposed method can follow the performance of the baseline algorithm in the so-called unstable environments in 2M time steps.

---

> ### Author Response · Authors · 2023-11-21
> **Respond to Questions**
>
> > What is the value of the hyper-parameter $t_{inter}$? How does its value affect the performance?
>
> The $t_{inter}$ value is the number of warm-up episodes. We need state data to train a KMeans classifier and this state data is collected within the $t_{inter}$ episodes. Additionally, the classifier and the MCT$^2$ are reconstructed every $t_{inter}$ episode to overcome the influence of policy shifts. In this paper, this hyper-parameter is empirically set as 500.
>
> > The target value $y^t$ in vanilla QMIX is $y^t=r^t+\gamma \max_{a_{t+1}}Q_{tot}(s^{t+1},a^{t+1})$, which is related to $s^{t+1}$. Why is $y^t$ still related to $s^t$ in Eq. (3)?
>
> Sorry for the confusion caused by our mistakes. The $y^t$ is related to $s^{t+1}$ in Eq.(3) (currently Eq.(4) in the revised version). The revised formula is
> $$
> y^t=r^t + \gamma[\mathbb{1}(c^{t+1}=\phi(s^{t+1}))\cdot Q_{assem}^{t+1}(s^{t+1}) +(1-\mathbb{1}(c^{t+1}=\phi(s^{t+1})))\cdot Q_{tot}(\tau,a^{t+1})]
> $$
> and we have updated it in the paper.
>
> > Is there any theoretical basis to prove that $Q_{assem}^t$ is more accurate than the original $Q_{tot}$?
>
> In this paper, similar states belong to a prototype cluster with a value, so similar states share the same value. The $Q_{assem}^t$ is the average value of $Q_{tot}$. In such a way, the expectation of $Q_{assem}^t$ is the same as the expectation of original $Q_{tot}$. The $Q_{assem}^t$ provides lower variance.
>
> > I think that in some scenarios, the SPP variant may be more likely to fall into a local optimal solution. Suppose that in such a scenario, agents can easily access the state corresponding to the suboptimal solution, while the state corresponding to the global optimal solution is in the opposite direction and relatively difficult to access (for example, further away from the initial position of the agents). The SPP variant may directly give up early exploration and find it difficult to converge to the global optimal solution. Of course, the above issue can be alleviated by adjusting cucb, but this requires sufficient prior knowledge.
>
> Thank you very much for your comments, we would love to discuss this scenario with you. You say that states oriented to suboptimal solutions are very easy to reach while states oriented to global optimal solutions are very difficult to collect and that the vast majority of data collected by algorithms focusing on exploration in this scenario are suboptimal, and we believe that obtaining an accurate value estimation of the global optimal state that is distinguishable from the suboptimal data is also a very difficult task to solve in this case. Our approach includes an annealing-like process to target such scenarios, in addition to the possibility of tuning the parameters of the UCT to encourage exploration. During annealing the tree in $\text{MCT}^2$ is destroyed and reconstructed, at which point the MARL algorithm resumes single-step exploration.
>
> In the remaining time we would like to address the remaining concerns that you have that prevent you from accepting the paper. We would appreciate it if you could engage with us on the remaining concerns, as we want to address them.

---

> > ### Comment · Reviewer_3kTw · 2023-11-21
> > **I have raised my score to 5.**
> >
> > Thanks to authors for the reply. Most of my concerns have been solved. But I still hope to see the results in more representative multi-agent domains, such as Google Research Football, and some other environments that are not Dec-POMDPs. After all, the trembling hand is a concept in general multi-agent games. And I think this paper still has a lot of room for improvement.
> >
> > I have improved my score, but I think the contribution of SPP still needs more convincing experimental results to support it.

---

> > > ### Author Response · Authors · 2023-11-23
> > > **Partial experimental results on Google Research Football**
> > >
> > > Thank you for your reply and raising the score. During this time, we also tested our approach on four academic scenarios from the Google Football environment against the QMIX algorithm. We do not have sufficient time to test many algorithms. The results are the average test scores among 5 seeds and are shown below:
> > >
> > > | counterattack_easy | 0.5M   | 1M     | 1.5M   | 2M     |
> > > | ------------------ | ------ | ------ | ------ | ------ |
> > > | SPP+QMIX           | 0.0014 | 0.0292 | 0.0675 | 0.1367 |
> > > | QMIX               | 0.0008 | 0.0085 | 0.0178 | 0.0178 |
> > >
> > > | counterattack_hard | 0.5M    | 1M      | 1.5M    | 2M     |
> > > | ------------------ | ------- | ------- | ------- | ------ |
> > > | SPP+QMIX           | -0.0112 | 0.0190  | 0.0226  | 0.0690 |
> > > | QMIX               | -0.0005 | -0.0017 | -0.0001 | 0.0193 |
> > >
> > > | 3v1_with_keeper | 0.5M   | 1M     | 1.5M   | 2M     |
> > > | --------------- | ------ | ------ | ------ | ------ |
> > > | SPP+QMIX        | 0.0460 | 0.0743 | 0.1118 | 0.1827 |
> > > | QMIX            | 0.0288 | 0.0547 | 0.0449 | 0.1047 |
> > >
> > > | corner   | 0.5M   | 1M     | 1.5M   | 2M     |
> > > | -------- | ------ | ------ | ------ | ------ |
> > > | SPP+QMIX | 0.0178 | 0.0233 | 0.0500 | 0.0668 |
> > > | QMIX     | 0.0103 | 0.0205 | 0.0150 | 0.0232 |
> > >
> > > In the remaining time, we would like to address the remaining concerns that you have that prevent you from accepting the paper. We would appreciate it if you could engage with us on the remaining concerns, as we want to address them.

---

### Official Review · Reviewer_RkkR · 2023-11-10

**Soundness:** 2 fair
**Presentation:** 3 good
**Contribution:** 2 fair
**Rating:** 5
**Confidence:** 3

**Summary:**

In order to balance between exploration and exploitation during the training process, the authors encourage the policy to follow the optimal trajectory as planned by a Monte-Carlo Trajectory Tree (MCT²). The MCT² is built upon historical trajectories, wherein states are organized into clusters via KMeans clustering. Within the MCT² framework, state values within the same cluster node are concurrently updated. The authors leverage PUCB values to find the optimal path across these clusters. During the rollout, when the actual state (cluster) diverges from the predicted state (cluster), the policy adopts an ε-greedy approach to facilitate exploration.
Experiments conducted within the SMAC benchmark show that the proposed method accelerates training and can be integrated into various MARL algorithms, including QMIX, QPLEX, and OW_QMIX.

**Strengths:**

The authors innovatively apply Monte-Carlo Tree structure into MARL context, leading to increased training speed. The proposed method may be applied to various existing MARL algorithms, thereby potentially contributes to the field of MARL research.

**Weaknesses:**

The experiment results do not conclusively demonstrate the effectiveness of the proposed method. In Figure 8, the performance of the proposed policy closely mirrors that of the original QMIX implementation. I would suggest the authors to test on more challenging MARL benchmarks, though those benchmarks often require more exploration, which may pose challenges for the proposed method.

Also, many MARL algorithms already suffer from a lack of exploration. The proposed method, in its pursuit of faster convergence, makes the additional trade-off of further diminishing exploration in favor of exploitation. This strategy necessitates careful consideration due to the potential consequences it may have on the algorithm's overall effectiveness.

**Questions:**

- In Section 6, the authors claim that the proposed method can be applied to the critic training in Actor-Critic MARL alrogithms. Can you briefly describe how to implement the proposed method in, say, MAPPO? And what is the performance improvement when applying to MAPPO?
- In the matrix game presented in Section 1, should the $epsilon$ for player 1 be 0.1?
- Can the proposed method be applied to scenarios with continuous action spaces?

---

> ### Author Response · Authors · 2023-11-21
> **Respond to Weaknesses**
>
> We thank the reviewer for the detailed review of our paper. After reading the reviews, we address your concerns below and update a revision of our paper. Please let us know if further clarification is needed.
>
> > In Figure 8, the performance of the proposed policy closely mirrors that of the original QMIX implementation. I would suggest the authors test on more challenging MARL benchmarks.
>
> In this paper, we provide our main experimental results in Figure 5 which contains the challenging subtasks of the SMAC environment. Among the subtasks, the 5m\_vs\_6m, 3s\_vs\_5z, and 2c\_vs\_64zg tasks are hard tasks. Not all the baseline algorithms can achieve good results. Additionally, the subtasks including MMM2, 3s5z\_vs\_3s6z, and 6h\_vs\_8z are super-hard tasks. Experimental results on these tasks within 2M time steps can show the better performance of our approach. In contrast, Figure 8 shows the other tasks in the SMAC environment and most of them are easy tasks in which the original QMIX as well as other baseline algorithms can achieve acceptable results. This might be the reason why our proposed policy closely mirrors that of the original QMIX.
>
> > The proposed method, in its pursuit of faster convergence, makes the additional trade-off of further diminishing exploration in favor of exploitation.
>
> Our method suggests dividing the decision-making of the existing MARL methods into two phases: our Stable Prefix Policy and vanilla policy. SPP balances the exploration and exploitation during the trajectory planning process with UCT, and we further apply Dirichlet noise to the planning phase, which gives more explorations. SPP explores more globally and centrally compared to existing popular MARL methods.

---

> ### Author Response · Authors · 2023-11-21
> **Resond to Questions**
>
> > In Section 6, the authors claim that the proposed method can be applied to the critic training in Actor-Critic MARL algorithms. Can you briefly describe how to implement the proposed method in, say, MAPPO? And what is the performance improvement when applying to MAPPO?
>
> In the Actor-Critic MARL algorithms, actor networks are responsible for training the action distributions conditioned on observations and the critic networks train the values of states. Applying our SPP method to the Actor-Critic method includes (i)assembling the values with the same cluster nodes and (ii)removing the actors' explorations by taking the maximum of the distributions and using the trajectory templates generated by the planning process to determine whether to interrupt the process and resume single-step explorations like non-SPP-augmented methods. Our approach contributes to the performance improvement of the Actor-Critic method in two ways: on the one hand, assembles the values with the same cluster nodes so that the values for training are more accurate; on the other hand, it determines whether the elimination of the exploration of the first half of the strategy can lead to more fine-grained improvements in the second half of the strategy by planning, which has already balanced exploration and exploitation by UCT.
>
> To focus on the efficiency aspects, we chose three subtasks to show the performance of the MAPPO and our SPP-augmented MAPPO within 2M time steps. The experimental results indicate that our method can improve the performance of Actor-Critic MARL methods. The items in the table are the average winning rates (%) among 5 seeds and their variances.
>
> | MMM2      | 0.5M          | 1M              | 1.5M            | 2M              |
> | --------- | ------------- | --------------- | --------------- | --------------- |
> | MAPPO     | 4.46$\pm$9.97 | 17.27$\pm$30.11 | 34.27$\pm$35.44 | 42.65$\pm$37.72 |
> | SPP+MAPPO | 9.34$\pm$5.54 | 46.81$\pm$19.63 | 51.03$\pm$14.95 | 62.42$\pm$14.98 |
>
> | 5m_vs_6m  | 0.5M           | 1M              | 1.5M            | 2M              |
> | --------- | -------------- | --------------- | --------------- | --------------- |
> | MAPPO     | 6.57$\pm$1.47  | 17.83$\pm$16.47 | 45.38$\pm$15.48 | 49.58$\pm$23.32 |
> | SPP+MAPPO | 36.58$\pm$6.54 | 53.32$\pm$11.27 | 57.57$\pm$11.19 | 60.65$\pm$10.56 |
>
> | 3s5z_vs_3s6z | 0.5M          | 1M             | 1.5M            | 2M              |
> | ------------ | ------------- | -------------- | --------------- | --------------- |
> | MAPPO        | 8.00$\pm$1.79 | 1.96$\pm$3.92  | 9.17$\pm$17.84  | 13.09$\pm$21.71 |
> | SPP+MAPPO    | 5.63$\pm$0.97 | 7.29$\pm$12.63 | 15.02$\pm$26.02 | 20.80$\pm$35.85 |
>
>
>
>
>
> > In the matrix game presented in Section 1, should the epsilon for player 1 be 0.1?
>
> Your concerns are much appreciated. We double-check the corresponding section. With $\epsilon$ set to 0.2, the player will have a 0.8 probability of going into the greedy mode and a 0.2 probability of going into a mode where the action is chosen randomly; however, since there is also a 0.1 probability that the original greedy action will be chosen in random mode, the value of $\epsilon$ should be taken as 0.2, at which point the greedy action will take the value of 0.1
>
> > Can the proposed method be applied to scenarios with continuous action spaces?
>
> The original intent of our paper was to address the problem of trembling hands that exists in value-based MARL methods. However, our SPP method contains major components: the UCT planning[1], Q-value assembling and cancelation of action noise are technically applicable to continuous action spaces. We have found that our approach is surprisingly able to improve the performance of Actor-critic based, so we are reasonably confident that our SPP approach is also capable of applying to continuous action spaces. We will actively search for suitable benchmarks for testing and welcome your further suggestions.
>
> In the remaining time we would like to address the remaining concerns that you have that prevent you from accepting the paper. We would appreciate it if you could engage with us on the remaining concerns, as we want to address them.
>
>
>
> [1] Hubert, Thomas, et al. "Learning and planning in complex action spaces." International Conference on Machine Learning.

---

> > ### Comment · Reviewer_RkkR · 2023-11-23
> >
> > Sorry for the late reply. I appreciate authors for the detailed response addressing my concerns. However, there are a few additional queries I have regarding the paper.
> >
> > - Final Performance Comparison. While the use of SPP appears to accelerate training during the early phase, I am interested in the final performance of MARL algorithms with and without SPP. Can you provide the final performance comparison of QMIX and MAPPO on SMAC?
> > - SPP's Performance Beyond SMAC Scenarios. Your experiments suggest that most scenarios in SMAC are not challenging enough and therefore weaken your results. Can you test SPP in other environments?
> > - Your response mentioned that "SPP explores more globally and centrally compared to existing popular MARL methods." Could you further explain how it explores globally and centrally at the same time?
> >
> > Overall, I would like to keep my score as it stands now.

---

> > > ### Author Response · Authors · 2023-11-23
> > > **Additional results and answers**
> > >
> > > We thank the reviewer for the detailed review of our paper and the reply. Based on the three points, we will try our best to address your concerns.
> > >
> > > > Final Performance Comparison. While the use of SPP appears to accelerate training during the early phase, I am interested in the final performance of MARL algorithms with and without SPP. Can you provide the final performance comparison of QMIX and MAPPO on SMAC?
> > >
> > > We have provided the final performances within 2M time steps in the appendix of our paper. Most of the easy tasks and hard tasks have already converged to near-optimal results and reached nearly 100% winning rates in later time steps. Thus we select some super-hard scenarios and test our approach against QMIX and MAPPO within 10M time steps. Due to the time limit and the new MAPPO integration, we only have time to finish the experiment of SPP+QMIX. The items in the table are the average value among 5 seeds.
> > >
> > > |              | SPP+QMIX | QMIX  | MAPPO |
> > > | ------------ | -------- | ----- | ----- |
> > > | 3s5z_vs_3s6z | 0.928    | 0.847 | 0.894 |
> > > | corridor     | 0.997    | 0.851 | 0.991 |
> > > | 5m_vs_6m     | 0.905    | 0.841 | 0.915 |
> > > | 27m_vs_30m   | 0.999    | 0.998 | 0.953 |
> > > | 6h_vs_8z     | 0.873    | 0.842 | 0.903 |
> > > | MMM2         | 0.962    | 0.875 | 0.921 |
> > >
> > > > SPP's Performance Beyond SMAC Scenarios. Your experiments suggest that most scenarios in SMAC are not challenging enough and therefore weaken your results. Can you test SPP in other environments?
> > >
> > > During this time, we also tested our approach on four academic scenarios from the Google Football environment against the QMIX algorithm. We do not have sufficient time to test many algorithms. The results are the average test scores among 5 seeds and are shown below:
> > >
> > > | counterattack_easy | 0.5M   | 1M     | 1.5M   | 2M     |
> > > | ------------------ | ------ | ------ | ------ | ------ |
> > > | SPP+QMIX           | 0.0014 | 0.0292 | 0.0675 | 0.1367 |
> > > | QMIX               | 0.0008 | 0.0085 | 0.0178 | 0.0178 |
> > >
> > > | counterattack_hard | 0.5M    | 1M      | 1.5M    | 2M     |
> > > | ------------------ | ------- | ------- | ------- | ------ |
> > > | SPP+QMIX           | -0.0112 | 0.0190  | 0.0226  | 0.0690 |
> > > | QMIX               | -0.0005 | -0.0017 | -0.0001 | 0.0193 |
> > >
> > > | 3v1_with_keeper | 0.5M   | 1M     | 1.5M   | 2M     |
> > > | --------------- | ------ | ------ | ------ | ------ |
> > > | SPP+QMIX        | 0.0460 | 0.0743 | 0.1118 | 0.1827 |
> > > | QMIX            | 0.0288 | 0.0547 | 0.0449 | 0.1047 |
> > >
> > > | corner   | 0.5M   | 1M     | 1.5M   | 2M     |
> > > | -------- | ------ | ------ | ------ | ------ |
> > > | SPP+QMIX | 0.0178 | 0.0233 | 0.0500 | 0.0668 |
> > > | QMIX     | 0.0103 | 0.0205 | 0.0150 | 0.0232 |
> > >
> > > > Your response mentioned that "SPP explores more globally and centrally compared to existing popular MARL methods." Could you further explain how it explores globally and centrally at the same time?
> > >
> > > Our method suggests dividing the decision-making of the existing MARL methods into two phases: our Stable Prefix Policy and vanilla policy. SPP balances the exploration and exploitation during the trajectory planning process with UCT. 1) We sample a large number of trajectories to construct the MCT$^2$, so the UCT planning is based on most of the transitions in the replay buffer. Compared with TD updates within a mini-batch of trajectories, planning through the replay buffer provides more **global** explorations. 2) The planning process does not consider the distributed execution and the credit allocation problem in MARL. Our method uses the global state or the observation concatenation of all the agents to perform feature abstraction for each step of the decision planning, which is more **central** compared to the existing MARL approaches.
> > >
> > >
> > >
> > > In the remaining time, we would like to address the remaining concerns that you have that prevent you from accepting the paper. We would appreciate it if you could engage with us on the remaining concerns, as we want to address them.

---

### Meta-Review · Area_Chair_JuQz · 2023-12-06

**Metareview:**

This paper studies the exploration-exploitation tradeoff in multi-agent reinforcement learning, by proposing a new method of constructing some Stable Prefix Policy using Monte-Carlo Trajectory Tree. The paper studies an interesting problem, with new algorithmic ideas. However, it reaches a consensus that the paper could be improved by providing more effective and sufficient experimental results, in terms of comparisons with baselines, and testing the settings beyond the cooperative ones. The theoretical results and writing may use some improvement also. Thus, I suggest the authors incorporate all the feedback and resubmit to some upcoming ML venues.

**Justification For Why Not Higher Score:**

The paper has a number of aspects that might need to be further improved. The rebuttals have not fully addressed all the comments from the reviewers.

**Justification For Why Not Lower Score:**

N/A

---

### Decision · Program_Chairs · 2024-01-16

Reject